# Advancing High-Throughput Phenotyping of Wheat in Early Selection Cycles

**Yuncai Hu, Samuel Knapp** and **Urs Schmidhalter** *

Chair of Plant Nutrition, Department of Plant Science, Technical University of Munich, D-85354 Freising, Germany; hu@wzw.tum.de (Y.H.); knapp@wzw.tum.de (S.K.)
* Correspondence: schmidhalter@wzw.tum.de; Tel.: +49-8161-713390

**Abstract:** Enhancing plant breeding to ensure global food security requires new technologies. For wheat phenotyping, only limited seeds and resources are available in early selection cycles. This forces breeders to use small plots with single or multiple row plots in order to include the maximum number of genotypes/lines for their assessment. High-throughput phenotyping through remote sensing may meet the requirements for the phenotyping of thousands of genotypes grown in small plots in early selection cycles. Therefore, the aim of this study was to compare the performance of an unmanned aerial vehicle (UAV) for assessing the grain yield of wheat genotypes in different row numbers per plot in the early selection cycles with ground-based spectral sensing. A field experiment consisting of 32 wheat genotypes with four plot designs (1, 2, 3, and 12 rows per plot) was conducted. Near infrared (NIR)-based spectral indices showed significant correlations ($p < 0.01$) with the grain yield at flowering to grain filling, regardless of row numbers, indicating the potential of spectral indices as indirect selection traits for the wheat grain yield. Compared with terrestrial sensing, aerial-based sensing from UAV showed consistently higher levels of association with the grain yield, indicating that an increased precision may be obtained and is expected to increase the efficiency of high-throughput phenotyping in large-scale plant breeding programs. Our results suggest that high-throughput sensing from UAV may become a convenient and efficient tool for breeders to promote a more efficient selection of improved genotypes in early selection cycles. Such new information may support the calibration of genomic information by providing additional information on other complex traits, which can be ascertained by spectral sensing.

**Keywords:** genotypes; high-throughput; hyper- and multi-spectral sensors; phenotyping; phenomics; plant breeding; proximal sensing; row number; UAV

## 1. Introduction

As breeding crops with a high yield and superior adaptability is vital to ensuring global food security, new technologies will enhance plant breeding to meet these challenges [1–3]. In contrast to recent progress in DNA marker assays and sequencing technologies that enable the high-throughput genotyping of many individual plants at a relatively low cost, phenotyping large numbers of genotypes and mapping populations in field trials is still laborious and expensive [4]. Therefore, the current bottleneck in plant breeding research is phenotyping.

High-throughput phenotyping through the application of remote or proximal sensing that is currently a new frontier offers a rapid and non-destructive approach to plant phenotyping. Numerous studies have shown that the grain yield of wheat genotypes/lines can reliably be assessed by spectral sensing [5–8]. These studies have also demonstrated that high-throughput phenotyping from ground-based sensing could not only contribute to savings in time and costs, but also allow for more objective information and even re-assessments in later selection cycles because the objective

digital data collection can be permanently stored. More importantly, the availability of unmanned aerial vehicles (UAV) has rapidly increased in recent years [9–11]. The aerial platforms have an advantage over ground-based sensing platforms in generating surface maps in real time and measuring plant parameters from large numbers of plots at a time [11–13]. Using high-resolution and low-altitude UAVs can overcome further limitations of ground-based sensing platforms, such as the non-simultaneous measurement of different plots, trafficability, row, and plot geometries requiring specific sensor configurations and vibrations resulting from uneven field surfaces [14]. However, there is still a lack of data comparing ground- and aerial-based sensing employed to phenotype wheat genotypes with high-throughput sensing.

For wheat phenotyping, limited seeds and resources in early selection cycles force breeders to use small plots with single or multiple row plots in order to include the maximum number of genotypes/lines for their assessment [15,16]. Additionally, due to large panels and technical obstacles, it is still difficult to determine the grain yield; thus, generally, promising lines are scored based on visual assessments alone. An efficient method for more objectively assessing a large number of lines in early selection cycles for the indirect detection of yield or yield-related traits could enhance the selection efficiency and save time and costs. Therefore, to meet the requirements for early selection cycles of wheat breeding, it is necessary to evaluate high-throughput phenotyping methods applied to estimate the grain yield or yield-related traits in small plots in early selection cycles. However, while much work to date has focused on evaluations in large-size plots, few reports have addressed the capacity for high-throughput phenotyping in improving the efficiency in selection in early generations in smaller plots with a couple of or single rows [16]. Although some field studies on spectral sensing employed to estimate the grain yield of wheat genotypes have been conducted in plots with different row numbers [5–7,17], a comparison of the performance of different spectral sensing approaches as an indirect selection tool for the grain yield of wheat genotypes in plots with single and multi-row designs has not yet been reported. Different row numbers in plots may lead to different soil coverage, affecting spectral sensing. Therefore, a comparison of varying row designs in plots is required to evaluate the performance of spectral sensing in breeding nurseries with different row numbers. Our previous study compared proximal spectral sensing in field trials with single-, two-, three-, and four-row designs [17], but only a single wheat variety was tested. Furthermore, there is the possibility to reduce the bare soil coverage with UAV imagery by aiming to separate soil from plant pixels, which tends to be more important in using spectral sensing for single-row plots that may also have greater row spacing between plots. Therefore, we hypothesized that aerial-based sensing from UAV could be more suitable for fewer rows or small plots.

In the present study, the objectives were to compare the performance of ground-based hyperspectral and aerial-based multispectral sensing from UAV for phenotyping the grain yield of 32 wheat genotypes/lines in 1-, 2-, and 3-row plots simulating the trial design for early selection cycles, and to compare the results with those of a 12-row plot design that is commonly used to evaluate the yield performance in breeding and agronomical evaluations in Western Europe. To the best of our knowledge, this study is the first to report on the use of both ground- and aerial-based spectral sensing (UAV) and to compare their performance for phenotyping the wheat grain yield in single-row plots with multi-row plots. The expected results may contribute to the further application of the spectral sensing technique in the program of early selection cycles in breeding.

## 2. Materials and Methods

### 2.1. Plant Material, Experimental Design, and Grain Yield Determination

Field experiments were conducted at the Dürnast Research Station of the Technical University of Munich in Germany (48°23′60″ N, 11°41′60″ E). The soil is a homogeneous Cambisol with a silty-clay loam texture. Annual precipitation is approximately 800 mm and the average temperature is 8 °C.

A randomized block design was used to test the phenotypic variation of 32 winter wheat cultivars (*Triticum aestivum* L.), including a panel of 26 modern wheat cultivars from Germany and six from Eastern Europe, with four plot designs that consisted of plots with 1, 2, 3, and 12 rows per plot and four replications. The space between plots was 50 cm and the space between rows was 12.5 cm, in order to achieve a high canopy coverage [17]. The length of the plots was increased to 10 m to further evaluate the influence of plot length on spectral measurements by subdividing them into different plot lengths in order to compare the effect of plot length on spectral measurements. Since there was no difference in spectral measurements due to the plot length from 2 to 10 m, the results are not presented in this paper. The plots were oriented from East to West.

At plant maturity, the grain yield was determined by the mechanical harvesting of 1-, 2-, 3-, and 12-row plots.

## 2.2. Spectral Reflectance Measurements Obtained by Ground-Based Hyperspectral Sensing and Aerial-Based Multispectral Sensing

In order to compare ground- and aerial-based sensing, a hand-held hyperspectral sensor and an unmanned aerial vehicle (UAV) carrying a multispectral camera were used for data acquisition at different BBCH scale (Biologische Bundesanstalt, Bundessortenamt und Chemische Industrie) growth stages [18], i.e., BBCH 49 (booting; 28.05.2018), BBCH 65 (anthesis; 10.06.2018), and BBCH 85 (grain filling; 26.06.2018). These stages were chosen due to being most indicative for the assessment of the winter wheat yield [19].

Ground-based hyperspectral sensing of the crop canopy was conducted through measurements of the reflected radiation. For measuring crop canopies, the primary focus was given to electromagnetic radiation within the visible (VIS, approximately 400–700 nm) and near-infrared (NIR, approximately 700–1100 nm) spectral range. A passive hand-held reflectance sensor (tec5, Oberursel, Germany) enabling hyperspectral readings was used. The bi-directional radiometer has a spectral detection range of 400 to 1100 nm and a bandwidth of 3.3 nm [20]. On clear sunny days at solar noon, canopy eflectance was measured at 0.5 m above the canopy with a 22° field of view (FOV) of a circular shape, resulting in about a 0.13 $m^2$ scanning area. The sensor was always positioned amid the row variants.

For aerial-based multispectral sensing, the wing aircraft senseFly ("eBee," SenseFly, Lausanne, Switzerland), equipped with a multispectral and sunshine sensor (Sequoia camera) (Parrot, Paris, France), was used for flying in an East-West direction. Since the Sequoia and sunshine sensor are integrated in a fixed structure, and imagery captured with the Parrot Sequoia camera is automatically recognized by the software, they are placed very accurately and the angles will not change at any time during the flight. The Sequoia multispectral camera takes photos in four spectral bands, i.e., green (550 nm, ~40 nm bandwidth), red (660 nm, ~40 nm bandwidth), and 2 x NIR regions (NIR-1: 735 nm, ~10 nm bandwidth and NIR-2: 790 nm, ~40 nm bandwidth) of the electromagnetic spectrum. The focal length is 3.98 mm; image size is 1280 x 960 pixels; and field of view is 61.9°horizontally, 48.5° vertically, and 73.7° diagonally. As the radiometric calibration target, a white balance card was used to enable the Pix4D software to calibrate and correct the images' reflectance. Flights were conducted at 50 m above the ground level, resulting in ground sampling distances of about 5 cm/pixel. Mission planning was done with eMotion 3 for the Sequoia camera. All flights were planned for 80% overlap along flight corridors and concomitantly carried out for the terrestrial sensor measurements at solar noon. Global shutters were used. The Pix4DMapper was used to process the multi-spectral UAV data. Plot-level means of green, red, NIR-1, and NIR-2 measurements from UAVs were created in ArcGIS Desktop version 10.5 (ESRI, Munich, Germany). To precisely extract the canopy coverage from individual plots, the shape files containing annotated single polygons with an optimized width to cover the most indicative sections of the row variants were segmented by hand. The width of the polygons was 15, 30, 40, and 110 cm for the 1-, 2-, 3-, and 12-row plots, respectively, to maximize the canopy coverage while minimizing bare-soil cover. For all flights, the GeoTIFFs with the green, red, NIR-1, and NIR-2

orthomosaics from Pix4D were combined with the plot polygon and shape file. Green, red, NIR-1, and NIR-2 means from each plot were generated using the zonal-statistics function in ArcGIS.

## 2.3. Calculation of Spectral Indices

Canopy spectral reflectance acquired from ground- and aerial-based spectral sensing was used to calculate vegetation indices, i.e., NIR-based indices of the water band index (WBI) and NIR:NIR, and visible- and NIR-based indices of the red normalized difference vegetation index (NDVI), NIR:Red, and NIR:Green (Table 1), which are reportedly highly correlated with the plant biomass and grain yield [19,21–23].

**Table 1.** Spectral reflectance indices used in this study.

| Index Name | Formula | | References |
|---|---|---|---|
| | **Ground-Based Hyperspectral Sensing** | **Aerial-Based Multispectral Sensing** | |
| **WBI (Water band index)** | **R900/R970** | | [21,22] |
| **NDVI (Red Normalized difference vegetation index)** | (R800−R680)/(R800+R680) | (R790−R660)/(R790+R660) | [20–23] |
| **Simple ratios** | NIR:NIR R780/R740 | R790/R735 | [24] |
| | NIR:Green R780/R550 | R790/R550 | [20] |
| | NIR:Red R760/R670 | R790/R660 | [23] |

## 2.4. Statistical Analysis

Lme4 and Sommer packages from the R-program (www.R-project.org) were used for the analysis of variance (ANOVA) to differentiate among row treatments. Phenotypic and genetic correlation coefficients were estimated to reveal the association between spectral indices and the grain yield.

The genetic correlation coefficients between spectral indices and the grain yield were calculated by the following formula [25]:

$$r_g = (Cov_{XY})/\sqrt{(Var_X \cdot Var_Y)}, \tag{1}$$

where Var and Cov refer to the components of variance and covariance, respectively, and $X$ and $Y$ are the two variables.

Broad-sense heritabilities ($h^2$) were calculated on a mean basis according to Holland et al. [26]. Broad-sense heritability is the proportion of the phenotypic variance, which is explained by the genetic variance, and was estimated as follows:

$$h^2 = \sigma^2{}_g/(\sigma^2{}_g + \sigma^2{}_e/n), \tag{2}$$

where $\sigma_g$ and $\sigma_e$ are the genotypic and residual variance components, respectively, and *n* is the number of replicate blocks.

## 3. Results

### 3.1. Genotypic Variation in Plots with 1, 2, 3, and 12 Rows

Significant genotypic variation in the grain yield among the 32 wheat genotypes was found in all row variants (Table 2). The highest mean grain yield per row was obtained from single-row plots, and the lowest yield was obtained from 12-row plots. The results showed that there was a significant effect of row variants on the grain yield. Heritability of the grain yield increased with an increasing

number of rows per plot and ranged from 0.82 to 0.92. The results also demonstrated that plots with fewer rows showed higher standard deviations (SD).

**Table 2.** Minimum, maximum, mean ± SD, and heritability ($h^2$) of the grain yield in plots with 1, 2, 3 and 12 rows. Mean comparison of plot treatments from Tukey's HSD test indicated significant differences at $p < 0.001$.

| Grain Yield (g/row) | Row Number Variants | | | | | | | | | | | |
|---|---|---|---|---|---|---|---|---|---|---|---|---|
| | 1 | | | 2 | | | 3 | | | 12 | | |
| Min | 1364 | | | 965 | | | 839 | | | 757 | | |
| Max | 2369 | | | 1662 | | | 1286 | | | 1055 | | |
| Mean±SD | 1836a | ± | 257 | 1281b | ± | 179 | 1035c | ± | 122 | 891d | ± | 99 |
| $h^2$ | | 0.82 | | | 0.87 | | | 0.9 | | | 0.92 | |

### 3.2. Phenotypic and Genetic Correlations Between Spectral Indices and the Grain Yield in Different Row Variants

The NIR-based indices NIR:NIR and WBI showed strong and significant phenotypic correlations with the grain yield, regardless of the number of rows and growth stage (Table 3). At grain filling, significant correlations of NIR:Red, NIR:Green, and NDVI with the grain yield were found almost for all row variants.

**Table 3.** Phenotypic correlation between the grain yield and spectral indices from ground- and aerial-based sensing in different row variants at BBCH scale stages 49, 65, and 85.

| Row Number Variants | Spectral Indices | | | | | | | | | |
|---|---|---|---|---|---|---|---|---|---|---|
| | NIR:Red | | NIR:Green | | NDVI | | NIR:NIR | | WBI | |
| **Ground-based hyperspectral sensing** | | | | | | | | | | |
| **BBCH 49** | | | | | | | | | | |
| 1 | 0.21 | ns | 0.30 | ns | 0.20 | ns | 0.50 | ** | 0.42 | * |
| 2 | 0.26 | ns | 0.29 | ns | 0.29 | ns | 0.58 | ** | 0.51 | ** |
| 3 | 0.41 | * | 0.36 | * | 0.45 | ** | 0.49 | ** | 0.51 | ** |
| 12 | 0.31 | ns | 0.28 | ns | 0.24 | ns | 0.37 | * | 0.44 | * |
| **BBCH 65** | | | | | | | | | | |
| 1 | 0.24 | ns | 0.38 | * | 0.28 | ns | 0.55 | ** | 0.16 | ns |
| 2 | 0.30 | ns | 0.37 | * | 0.30 | ns | 0.66 | ** | 0.42 | * |
| 3 | 0.22 | ns | 0.24 | ns | 0.26 | ns | 0.56 | ** | 0.60 | ** |
| 12 | −0.02 | ns | 0.06 | ns | −0.04 | ns | 0.44 | * | 0.47 | ** |
| **BBCH 85** | | | | | | | | | | |
| 1 | 0.43 | * | 0.31 | ns | 0.37 | * | 0.36 | * | 0.47 | ** |
| 2 | 0.66 | ** | 0.55 | ** | 0.63 | ** | 0.65 | ** | 0.72 | ** |
| 3 | 0.55 | ** | 0.36 | * | 0.54 | ** | 0.47 | ** | 0.60 | ** |
| 12 | 0.45 | * | 0.25 | ns | 0.51 | ** | 0.38 | * | 0.64 | ** |
| **Aerial-based multispectral sensing** | | | | | | | | | | |
| **BBCH 49** | | | | | | | | | | |
| 1 | 0.42 | * | 0.48 | ** | 0.35 | * | 0.54 | ** | - | |
| 2 | 0.55 | ** | 0.61 | ** | 0.52 | ** | 0.71 | ** | - | |
| 3 | 0.59 | ** | 0.63 | ** | 0.60 | ** | 0.61 | ** | - | |
| 12 | 0.37 | * | 0.35 | ns | 0.33 | ns | 0.51 | ** | - | |
| **BBCH 65** | | | | | | | | | | |
| 1 | 0.33 | ns | 0.38 | * | 0.31 | ns | 0.58 | ** | - | |
| 2 | 0.40 | * | 0.54 | ** | 0.41 | * | 0.67 | ** | - | |
| 3 | 0.36 | * | 0.53 | ** | 0.38 | * | 0.72 | ** | - | |
| 12 | 0.27 | ns | 0.35 | ns | 0.27 | ns | 0.58 | ** | - | |
| **BBCH 85** | | | | | | | | | | |
| 1 | 0.44 | * | 0.40 | * | 0.44 | * | 0.40 | * | - | |
| 2 | 0.68 | ** | 0.60 | ** | 0.68 | ** | 0.64 | ** | - | |
| 3 | 0.62 | ** | 0.44 | * | 0.63 | ** | 0.57 | ** | - | |
| 12 | 0.55 | ** | 0.49 | ** | 0.59 | ** | 0.45 | * | - | |

ns = not significant; * = $p < 0.05$; ** = $p < 0.01$.

Compared with ground-based sensing, spectral indices derived from aerial-based sensing consistently showed higher levels of phenotypic association with the grain yield.

Genetic relationships between spectral indices and the grain yield were similar to phenotypic relationships; however, the coefficients of genetic correlation were sometimes higher than those of the phenotypic correlations (Table 4).

**Table 4.** Genetic correlation between the grain yield and spectral indices from ground- and aerial-based sensing in different row variants at BBCH scale stages 49, 65, and 85.

| Row Number Variants | Spectral Indices | | | | | | | | | |
|---|---|---|---|---|---|---|---|---|---|---|
| | NIR:Red | | NIR:Green | | NDVI | | NIR:NIR | | WBI | |
| **Ground-based hyperspectral sensing** | | | | | | | | | | |
| **BBCH 49** | | | | | | | | | | |
| 1 | 0.44 | ns | 0.56 | ns | 0.37 | ns | 0.68 | * | 0.66 | * |
| 2 | 0.44 | ns | 0.50 | ns | 0.49 | ns | 0.80 | * | 0.56 | * |
| 3 | 0.54 | * | 0.53 | ns | 0.65 | * | 0.54 | * | 0.58 | * |
| 12 | 0.41 | ns | 0.37 | ns | 0.37 | ns | 0.40 | ns | 0.44 | * |
| **BBCH 65** | | | | | | | | | | |
| 1 | 0.65 | ns | 0.86 | * | 0.70 | ns | 0.75 | ** | 0.33 | ns |
| 2 | 0.48 | ns | 0.65 | * | 0.57 | ns | 0.81 | ** | 0.53 | * |
| 3 | 0.37 | ns | 0.39 | ns | 0.48 | ns | 0.71 | ** | 0.80 | ** |
| 12 | 0.01 | ns | 0.10 | ns | 0.02 | ns | 0.46 | * | 0.51 | * |
| **BBCH 85** | | | | | | | | | | |
| 1 | 0.51 | * | 0.41 | ns | 0.46 | ns | 0.40 | ns | 0.49 | * |
| 2 | 0.74 | ** | 0.67 | * | 0.70 | ** | 0.71 | ** | 0.75 | ** |
| 3 | 0.64 | ** | 0.50 | * | 0.64 | ** | 0.56 | * | 0.45 | ns |
| 12 | 0.48 | * | 0.28 | ns | 0.55 | * | 0.42 | * | 0.65 | ** |
| **Aerial-based multispectral sensing** | | | | | | | | | | |
| **BBCH 49** | | | | | | | | | | |
| 1 | 0.79 | * | 0.94 | ** | 0.57 | * | 1.00 | ** | - | |
| 2 | 0.75 | ** | 0.92 | ** | 0.74 | ** | 1.00 | ** | - | |
| 3 | 0.92 | * | 0.85 | ** | 0.77 | ** | 0.79 | ** | - | |
| 12 | 0.39 | ns | 0.36 | ns | 0.36 | ns | 0.55 | * | - | |
| **BBCH 65** | | | | | | | | | | |
| 1 | 0.71 | * | 0.64 | * | 0.59 | * | 0.87 | * | - | |
| 2 | 0.71 | * | 0.80 | ** | 0.67 | * | 0.79 | ** | - | |
| 3 | 0.53 | * | 0.69 | ** | 0.56 | * | 0.85 | ** | - | |
| 12 | 0.22 | ns | 0.37 | ns | 0.31 | ns | 0.61 | ** | - | |
| **BBCH 85** | | | | | | | | | | |
| 1 | 0.57 | * | 0.53 | * | 0.55 | * | 0.53 | * | - | |
| 2 | 0.74 | ** | 0.67 | ** | 0.74 | ** | 0.71 | ** | - | |
| 3 | 0.68 | ** | 0.51 | * | 0.70 | ** | 0.64 | ** | - | |
| 12 | 0.58 | ** | 0.53 | * | 0.63 | ** | 0.46 | * | - | |

ns = not significant; * = $p < 0.05$; ** = $p < 0.01$.

### 3.3. Heritability of Spectral Indices in Different Row Variants

A moderate to high level of broad-sense heritability was observed for most spectral indices. The heritability of spectral reflectance indices generally increased with the growth stage for a given row variant and with an increasing number of rows at any given growth stage (Table 5). The highest heritability for a given index was found at grain filling (BBCH 85) in 12-row plots.

For example, at grain filling, the heritability of indices in 12-row plots ranged from 0.65 to 0.96 for ground-based sensing, and from 0.79 to 0.96 for aerial-based sensing.

Compared with ground-based sensing, the values of heritability for the same index from aerial-based sensing were higher in most cases (Table 5).

**Table 5.** Heritability of spectral indices from ground- and aerial-based sensing in plots with different row variants at BBCH scale stages 49, 65, and 85.

| Row Number Variants | Spectral Indices | | | | |
|---|---|---|---|---|---|
| | NIR:Red | NIR:Green | NDVI | NIR:NIR | WBI |
| **Ground-based hyperspectral sensing** | | | | | |
| **BBCH 49** | | | | | |
| 1 | 0.51 | 0.52 | 0.52 | 0.64 | 0.65 |
| 2 | 0.49 | 0.44 | 0.39 | 0.49 | 0.57 |
| 3 | 0.57 | 0.50 | 0.54 | 0.72 | 0.76 |
| 12 | 0.58 | 0.55 | 0.52 | 0.86 | 0.88 |
| **BBCH 65** | | | | | |
| 1 | 0.36 | 0.37 | 0.33 | 0.67 | 0.80 |
| 2 | 0.63 | 0.47 | 0.48 | 0.72 | 0.82 |
| 3 | 0.55 | 0.56 | 0.41 | 0.70 | 0.81 |
| 12 | 0.50 | 0.52 | 0.55 | 0.89 | 0.92 |
| **BBCH 85** | | | | | |
| 1 | 0.80 | 0.65 | 0.77 | 0.76 | 0.81 |
| 2 | 0.85 | 0.69 | 0.84 | 0.78 | 0.79 |
| 3 | 0.84 | 0.66 | 0.84 | 0.81 | 0.80 |
| 12 | 0.96 | 0.88 | 0.95 | 0.96 | 0.95 |
| **Aerial-based multispectral sensing** | | | | | |
| **BBCH 49** | | | | | |
| 1 | 0.45 | 0.48 | 0.64 | 0.13 | - |
| 2 | 0.58 | 0.52 | 0.63 | 0.46 | - |
| 3 | 0.33 | 0.55 | 0.62 | 0.56 | - |
| 12 | 0.73 | 0.93 | 0.84 | 0.89 | - |
| **BBCH 65** | | | | | |
| 1 | 0.51 | 0.65 | 0.57 | 0.44 | - |
| 2 | 0.55 | 0.65 | 0.61 | 0.76 | - |
| 3 | 0.60 | 0.72 | 0.59 | 0.70 | - |
| 12 | 0.79 | 0.89 | 0.87 | 0.93 | - |
| **BBCH 85** | | | | | |
| 1 | 0.83 | 0.79 | 0.86 | 0.73 | - |
| 2 | 0.91 | 0.91 | 0.94 | 0.90 | - |
| 3 | 0.85 | 0.85 | 0.90 | 0.86 | - |
| 12 | 0.95 | 0.95 | 0.96 | 0.96 | - |

## 4. Discussion

### 4.1. Phenotypic and Genetic Correlations Between the Grain Yield and Spectral Indices, as Obtained from Ground-Based Hyperspectral and Aerial-Based Multi-Spectral Sensing

The evaluation and selection of moderate- and high-yielding wheat genotypes using spectral indices derived from ground-based sensing have been successfully applied under different environmental conditions in previous studies in plots with increased row numbers [5,6,8,27]. In the present study, the best performing spectral indices from ground-based hyperspectral sensing for predicting the grain yield at phenotypic and genetic levels were the NIR-based indices NIR:NIR and WBI for all variants with different numbers of rows per plot (Tables 3 and 4). Although the indices NIR:Red, NIR:Green, and NDVI could not distinguish among genotypes at BBCH 49 and BBCH 65, they significantly correlated with the grain yield at BBCH 85. Overall, these results agree with the findings presented by other authors studying wheat under well-watered and drought stress conditions [5–8,19].

This is the first report on a comparison of correlations between spectral indices and the wheat grain yield for plots with different numbers of rows. As single- or two-row plots share a higher fraction of soil coverage than plots with higher row numbers (3 or 12), this study aimed to assess whether differences in canopy/soil coverage representing mixed-pixel situations interfere with spectral sensing. A previous study [17] showed that the one-row design only covered approximately 34% of the field of view of a hand-held spectrometer, whereas two-row plots covered 80%, when the sensor was positioned at 100 cm above the plant canopy. To reduce the effect of bare soil between plots with one or two rows per plot, Barmeier and Schmidhalter [17] suggested optimizing spectral sensing by reducing

the sensor–canopy distance, with the sensor always being positioned amid rows. Therefore, the sensor–canopy distance was reduced from 100 to 50 cm in this study. By decreasing the sensor–canopy distance for a hand-held spectral proximal sensor, the results of this study showed that differences in grain yield among genotypes could not only be distinguished in multi-row plot designs, but in single-row plots as well, especially by NIR-based indices (Tables 3 and 4), thus suggesting that it is possible to use spectral sensing for the high-throughput phenotyping of wheat genotypes in early selection cycles. Reliable evaluation in smaller plots for the drivers of yield, especially those novel and rare alleles commonly lost when targeting the grain yield alone in breeder's nurseries, will enable novel phenotypes to be recycled through subsequent crossing and population development [16]. The new information may strongly support genomic selection efficiency, as well as the calibration of genomic information, by providing additional information on other complex traits, such as drought tolerance [8,28], salinity tolerance [29,30], and nitrogen use efficiency [19], which can be ascertained by spectral sensing.

In aerial-based multispectral sensing, significant correlations between spectral indices and the yield were generally higher than those obtained from ground-based sensing, especially at booting and anthesis (Tables 3 and 4), suggesting that increased precision may be obtained from UAV imagery. This is in agreement with recent reports [12,31,32]. The relatively higher precision of measurements by UAVs can be associated with several major factors: (i) Non-vegetation pixels can be better removed from imagery obtained by UAV. This could be more pronounced in plots with fewer rows. Recent studies have demonstrated the possibility to improve the segmentation of plant-soil pixels, e.g., using Support Vector Machine (SVM) classification or Convolutional Neural Networks [27,33,34]; (ii) aerial-based sensing has an advantage over ground-based sensing platforms in generating surface maps in real time and measuring plant parameters from a large number of plots at a time, typically associated with the time required to make ground-based measurements in large trials [12,13]; (iii) using high-resolution and low-altitude UAVs can overcome further limitations of ground-based sensing platforms, such as the non-simultaneous measurement of different plots, trafficability, row, and plot geometries requiring specific sensor configurations, and vibrations resulting from uneven field surfaces [12,28]. Given that the operation of UAV image acquisition is less labor-intensive, and owing to improved segmentation procedures and a higher precision than non-imaging proximal sensing, aerial-based multispectral sensing via UAV is expected to increase the efficiency of high-throughput phenotyping in large-scale plant breeding programs [10,12].

*4.2. Heritability of Spectral Indices from Ground- and Aerial-Based Sensing*

High heritability and strong phenotypic and genetic correlations between indirect traits and the grain yield are desirable [25]. However, in previous studies on wheat genotypes, heritability values of spectral indices have been inconsistent [5–8]. Falconer [25] proposed that using an alternative indirect selection trait for the grain yield is only appropriate if the heritability of the indirect trait is higher than that of the grain yield itself. Therefore, the authors [5–7] concluded that, even with low $h^2$ values, spectral indices can still be valuable as indirect selection traits to breeders because such values were still higher than those of the grain yield in most cases. The results of this study showed that compared with $h^2$ values of the grain yield, higher $h^2$ values of spectral indices were obtained, particularly from aerial-based sensing at grain filling in plots with 1, 2, and 12 rows (Tables 2 and 5), thus confirming that aerial-based sensing delivers a higher precision for high-throughput phenotyping.

## 5. Conclusions

Our study demonstrated that NIR-based spectral indices indicated strong and significant phenotypic and genetic correlations with the grain yield, regardless of the row variants per plot and growth stage, indicating a high potential of NIR-based indices as indirect selection traits for the wheat grain yield. Compared with ground-based sensing, spectral indices from aerial-based sensing by UAV consistently showed a higher association with the grain yield, indicating that an increased

precision may be obtained and is expected to increase the efficiency of high-throughput phenotyping in large-scale plant breeding programs and allow breeders a more objective selection of improved genotypes in early selection cycles, thereby reducing costs by performing fewer directed samplings.

**Author Contributions:** U.S. conceived and designed the experiments; Y.H. performed the experiments; Y.H. and S.K. analyzed the data; Y.H. and U.S. wrote the paper. All authors have read and agreed to the published version of the manuscript.

**Funding:** This research was funded by German research foundation (DFG) grant SCHM1456\8-1.

**Conflicts of Interest:** The authors declare no conflicts of interest.

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
