# Peer review of "Advancing High-Throughput Phenotyping of Wheat in Early Selection Cycles"

_remotesensing, doi:10.3390/rs12030574_

Round 1
Reviewer 1 Report
Advancing high‐throughput phenotyping of wheat in early selection cycles is about the study of phenotypic variation of 32 winter wheat . Different indices were used to build correlation between grain yield and growth stages. Overall study is good but there are some suggestions.
How ground truthing was done.
Abstract needs to be revised. See comments in the attached file.
Introduction needs to be written by building background in relation with the problem to be addressed.
Discussion needs to be improved.
Conclusion should be based on present work.

Reviewer 2 Report
Dear Authors
First of all, I will thank you for the paper and its clarity for a reader. High-throughput phenotyping is of great interest for farmers and breeders.
I have however some suggestions to improve or explain more in details what you have done.
In the abstract, I am a little bit surprised not to have the terms hyper or multi-spectral in the keywords ! Moreover, soil analysis could be added at the end of the abstract.
Line 71, you can probably give more information on the segmentation algorithms you cite, since it exists a lot of this kind of algorithms, which can be used with more or less success.
In §2.2, you mention hyperspectral information for ground-based sensor and multispectral sensor for aerial sensor. Why not to have used a hyperspectral system for aerial sensor ? I know this is expensive but it could better for the comparison. For the hyperspectral information, you explain that the sensor can be used until 1100 nm but the indices you used are limited to 970 nm.
Line 108, you did not mention any information on the illumination perturbations ? May you give also more details on the Sequoia camera ? How the bandwiths were chosen ?
I also didn't understand the legend of the table 1 concerning the animals or humans information ??!!
you may also numbered the equation presented and change the font lines 155-156.
Concerning the use of the sensors, you need to give more details on the calibration of the different sensors, especially for the grain-yield sensor used to have the ground truth.
Did you do some radiometric/geometric corrections for the UAV data ? How ?
finally I am not agree with you (line 220) when you indicate that the use of spectral sensing for high-throughput phenotyping is possible. It depends, to my opinion, on the kind of the crop ! Indeed, when you test spectral sensor on very small inter-row spacing, it seems difficult.
The conclusion is ok for me, except that your tests has been done for one genotype.
Reviewer 3 Report
The manuscript presents a study on the use of spectral images for high-throughput phenotyping, using both proximal and UAV-sensing. The article is well-written and of interest to the scientific community. I have only a few minor remarks:
- Although the manuscript is well-written as a whole, a full text revision is recommended to correct some minor problems that can be found throughout the text.
- Line 112: please explain the choice of using a fixed-wing UAV in the experiments. Rotating wing aircraft would seem to be a better choice in this case, as plots for phenotyping are usually relatively small. In addition, since the drones were used at relatively low altitudes (50 m), blur effects would tend to be smaller.
- Line 118: is such a level of overlapping really needed?
- Table 1: I think the sentence about ethical approval was included by mistake in the caption?
- Table 3: please indicate what “ns”, “*” and “**” mean.
- Lines 252-255: I think these sentences were included by mistake.
Round 2
Reviewer 1 Report
Article is now in good shape.
Author Response
Dear Reviewer,
Many thanks for your positive support.
Regards,
Urs Schmidhalter